# The Predictive Impact of HPV Genotypes, Tumor Suppressors and Local Immune Response in the Regression of Cervical Intraepithelial Neoplasia 2-3: A Prospective Population-Based Cohort Study

**DOI:** 10.3390/ijms26115205

**Published:** 2025-05-28

**Authors:** Pavla Sustova, Birgit Engesæter, Irene Tveiterås Øvestad, Einar G. Gudlaugsson, Reza Ghiasvand, Ivar Skaland, Jan P. A. Baak, Ameli Tropé, Emiel A. M. Janssen, Ane Cecilie Munk

**Affiliations:** 1Department of Laboratory Medicine, St. Olav’s Hospital, Trondheim University Hospital, 7006 Trondheim, Norway; 2Department of Clinical and Molecular Medicine, Faculty of Medicine and Health Sciences, Norwegian University of Science and Technology, 7030 Trondheim, Norway; 3Department of Cervical Cancer Screening, Cancer Registry of Norway, Norwegian Institute of Public Health, 0304 Oslo, Norway; 4Department of Pathology, Stavanger University Hospital, 4011 Stavanger, Norway; 5Oslo Centre for Biostatistics and Epidemiology, Oslo University Hospital, 0424 Oslo, Norway; 6Department of Research, Cancer Registry of Norway, Norwegian Institute of Public Health, 0304 Oslo, Norway; 7Department of Chemistry, Bioscience and Environmental Engineering, University of Stavanger, 4021 Stavanger, Norway; 8Department of Obstetrics and Gynecology, Sørlandet Hospital Kristiansand, 4604 Kristiansand, Norway

**Keywords:** cervical intraepithelial neoplasia, high-risk HPV, HPV genotypes, regression, biomarkers, CIN lesion size

## Abstract

Cervical intraepithelial neoplasia (CIN) is caused by human papillomavirus (HPV); however, factors such as HPV genotype and individual immune response may also contribute to its development. The loop electrosurgical excision procedure (LEEP) is a treatment for high-grade cervical intraepithelial neoplasia (CIN), as approximately 30% of these cases may progress to cancer. However, 20–40% of cases will regress spontaneously. HPV16 infection constitutes the highest risk for progression to cervical cancer and a lower probability of regression. Knowledge regarding the regression of lesions caused by other high-risk genotypes alone or in association with biomarker expression and lesion length has been limited. In the present study, the regression rates of high-grade squamous intraepithelial lesions were calculated. Twenty-one percent of the 161 women diagnosed with CIN2-3 on colposcopy-directed biopsies exhibited regression (defined as CIN1 or less) in the subsequent cone excisions. The mean interval between biopsy and treatment was 113 days (range of 71–171). High-grade lesions of the squamous epithelium caused by HPV16, together with lesions caused by HPV31, 52 and 58, showed significantly lower regression rates (HR 0.54, 0.22–0.75; low-regression group) than lesions caused by HPV18, 33, 35, 39, and 45 (HR 2.85, 1.54–5.28; high-regression group). A multivariate analysis of HPV genotypes, epithelial expressions of pRb and p53, immune cell proportions in the stroma (CD4/CD25 and CD4/CD8), and lesion lengths correctly predicted regression in 78% (Harrell’s C). A Harrell’s C value of 82% for the low-regression group indicates that different HPV genotypes or groups, together with divergent patterns of tumor suppressors, immune cells, and lesion size, can give prognostic information regarding the outcome of CIN2-3.

## 1. Introduction

High-risk human papillomavirus (hr-HPV) is the most common sexually transmitted agent worldwide and a necessary cause of almost all cervical intraepithelial neoplasia (CIN) [1,2] and 80–85% of adenocarcinomas in situ of the uterine cervix (AIS) [3,4]. Both premalignant conditions can lead to cancer development, with AIS appearing to have a shorter preinvasive phase than CIN [5]. Even though AIS seems to have higher oncogenic potential [6], most studies do not distinguish between these two subtypes when assessing potential biomarkers for the progression or regression of premalignant lesions [7]. The current study investigates prognostic biomarkers solely for CIN.

Most HPV infections clear within 1–2 years without any need for therapy [8,9]. However, 10–20% of infected women will have persistent infections with increased risk of developing high-grade dysplasia (CIN2-3) [10]. Approximately 30% of all CIN3 lesions will progress to cancer within 30 years [10], whereas 20–40% will regress spontaneously [11,12,13,14,15]. The current treatment involves surgical excision of cervical tissue [16,17], with potential side effects; the most serious is an increased relative risk (RR) of premature birth (<37 weeks) [18]. Several studies have explored the oncogenic potential of different HPV genotypes [19,20], but there is limited understanding of the regression rates associated with different genotypes. Enhancing our understanding of the regression potential and immune system impact of various HPV genotypes could greatly improve follow-up strategies for HPV-positive women.

The immune response is an important factor in overcoming an HPV infection. Differences regarding the number and composition of infiltrating immune cells have been observed in persistent CIN3 compared to regressive CIN3 lesions [12,13,21,22]. The immune system requires time for eliminating an HPV infection, making time a critical factor for both the regression and progression rates of CIN. Insinga et al. found that the cumulative risk of progression to CIN2-3 with persistent HPV16 infection was 6.9% after 12 months, 12.9% after 24 months, and 16.5% after 36 months [23].

To fulfil its viral lifecycle, HPV infects the basal layer of squamous epithelial cells in the cervical mucosa, where HPV oncogenes E6 and E7 inactivate both pRb and p53 [24,25]. The downregulation of pRb hampers the differentiation of epithelial cells, and the lower activity of p53 inhibits the natural growth arrest of infected cells. Consequently, the virus can utilize the host DNA replication machinery to produce viral DNA [21].

A previous publication from our prospective study demonstrated the impact of factors such as CIN length, pRb expression, and the proportion of CD4-positive cells in the stroma, alongside clinical factors like condom use, as prognostic markers for the spontaneous regression of CIN2-3 [12]. The aim of this part of the study was to evaluate the prognostic significance of different HPV genotypes and genotype groups, along with the intraepithelial expressions of tumor suppressor genes p53 and pRb, lesion size, and stromal immune cell subsets specifically CD4 (T-helper cells), CD8 (effector T-cells), and CD25 (interleukin-2) in predicting the clinical outcomes of high-grade squamous lesions (CIN2-3).

Identifying clinically useful prognostic markers is of great value for improving the risk stratification of CIN, reducing the over-treatment of regressing high-grade intraepithelial lesions, and ensuring efficient follow-up without unnecessary delays in treatment for patients at high risk of developing cancer.

## 2. Results

The study population consisted of 161 healthy women with first-time diagnoses of CIN2-3 in colposcopy-directed biopsies. The mean age was 31 years (range: 25–41) and the mean follow-up interval between biopsy and cone excision was 113 days (range: 84–171). Of the 161 women, 34 were diagnosed with CIN1 or normal histology in the following cone excision, giving an overall regression rate of 21%. There were no significant differences in mean age between the regression and persistent groups (31.1 years (25.1–41.0) versus 31.8 years (25.0–41.0), *p* = 0.15). None of the patients included had cervical cancer in the cone tissue. The results of pRb-positive cells in the lower half of the epithelium (pRbLH) and the CIN lesion lengths were presented in a former publication on the actual study population [12].

### 2.1. Genotypes and Regression Rate

The number of HPV genotypes detected for each patient ranged from one to five. Using a linear array (Roche), ninety-three patients tested positive for only one genotype. Fifty patients were positive for two, thirteen were positive for three, two were positive for four, and three were positive for five genotypes. In samples positive for more than one genotype, each genotype was included in the analyses separately. The regression rate for each HPV genotype is presented in Table 1. Compared to the regression rate calculated for the whole study population, HPV35 was the only genotype with a statistically significant higher regression rate (HR 2.48, 1.27–4.86). CIN lesions were subsequently categorized into two groups according to their association with regression rate. Squamous intraepithelial lesions positive for HPV18, 33, 35, 39, and 45 (*n* = 60) were almost three times more likely to regress (HR 2.85, 1.54–5.28) compared to lesions negative for those subtypes (Table 2), and were defined as the high-regression group. Lesions positive for HPV16, 31, 52, and 58 (*n* = 97) had a significantly lower regression rate (HR 0.54, 0.22–0.75) compared to lesions negative for those subtypes, and were defined as the low-regression group. Furthermore, the high-regression group showed a regression rate of 3.20 (2.09–4.30) per 1000 women compared to 1.18 (0.58–1.78) per 1000 women in the low-regression group (Table 2). By excluding those with more than one infection, we did not find any significant difference in the probability of regression (Appendix A). Seventeen women tested negative for all genotypes in both the low- and high-regression groups but were positive for one or more of eleven HPV genotypes categorized as “Other HPVs”. Each of these genotypes was detected in five or fewer women: HPV6 (*n* = 2), HPV42 (*n* = 1), HPV53 (*n* = 5), HPV54 (*n* = 1), HPV56 (*n* = 1), HPV59 (*n* = 1), HPV61 (*n* = 1), HPV66 (*n* = 2), HPV68 (*n* = 1), HPV73 (*n* = 1), and HPV82 (*n* = 1). Among these seventeen women, four experienced regression corresponding to a regression rate of 2.03 (0.29–3.77).

### 2.2. Levels of Biomarkers and CIN Lesion Length in Relation to Regression Rate

The association between the regression rate and the levels of tumor suppressors (pRb and p53), the presence and ratios of different immune cells (CD4, CD8, and CD25), and the lesion lengths were calculated for the whole study population. No relationship was found using continuous variables. However, using cut-off values calculated from the ROC curves that generated categorical data, lesions with ≤30% pRbLH showed a significantly lower probability of regression (7% (4/62)) regardless of genotype association, compared to lesions with >30% pRbLH, which showed a regression rate of 31% (30/98) (HR 0.21, 0.07–0.61) (Table 3). Likewise, lesions with >10% of cells positive for p53 in the lower half of the epithelium (p53LH) were associated with a lower regression rate of 14% (11/79) compared to lesions with ≤10% p53LH (regression rate: 28% (23/81); HR 0.32, 0.15–0.71). A regression rate of 12% (12/99) was observed among women with lesion lengths >2.5 mm compared to 35% (22/62) in women with smaller lesions ≤ 2.5 mm (HR 0.39, 0.19–0.80) (Table 3).

Neither the absolute number of CD4-, CD8-, CD25-, and CD138-positive cells in the epithelium or stroma nor the correlating categorical values had a significant association with regression (Appendix A). However, calculating the ratios between CD4/CD25 and CD4/CD8 in the stroma revealed significant differences. A CD4/CD25 ratio >9.75 increased the probability of regression to 39% (7/18) compared to 18% (26/141) in those with a CD4/CD25 ratio ≤ 9.75 (HR 4.96, 2.46–9.99), while CD4/CD8 > 0.67 had a significantly lower regression rate of 12% (6/52) compared to a regression rate of 25% (27/108) for a ratio ≤ 0.67 (HR 0.43, 0.21–0.91).

### 2.3. Levels of Biomarkers, CIN Lesion Lengths, and Regression Rates Relative to HPV Genotypes

Using multivariate analysis, both the high- and low-regression groups showed the same tendency as for the whole study population regarding regression related to biomarker expression and lesion length. However, the associations were strengthened, as summarized in Table 3. In the low-regression group, only 1/46 (2%) of the lesions regressed if pRbLH ≤ 30%, compared to 12/50 (24%) when pRbLH > 30% (HR 0.09, 0.01–0.72). Similarly, lesions in the low-regression group with p53LH > 10% revealed a regression rate of 9% (4/46) versus 18% (9/50) in lesions when p53LH ≤ 10% (HR 0.20, CI: 0.04–0.88). The stromal ratio of CD4/CD25 in the low-regression group showed that a higher ratio (>9.75) was associated with an even significantly higher difference in a regression rate of 21% (3/14), versus 12% in those with a ratio ≤ 9.75 (10/82) (HR 6.17, 1.29–29.57). In the high-regression group, only a CD4/CD25 ratio of >9.75 significantly predicted regression (HR 4.27, 1.22–14.87).

Harrell’s C risk score, based on the expressions of pRb and p53, lesion length, and the CD25/CD4 and CD4/CD8 ratios in the stroma, correctly predicted regression in 82% of the patients in the low-regression group, 71% in the high-regression group, and 78% for the study population in total.

The CD25str, CD25epi, CD4str, CD4epi, CD8str, CD8epi, CD138str, CD138epi, CD4str/CD8str, CD4epi/CD8epi, CD25str/CD8str, and CD25epi/CD8epi patterns were also analyzed in the low-regression group compared to HPV35 lesions, and in the high-regression group compared to HPV35 lesions; however, no significant differences were found.

## 3. Discussion

The natural history of CIN- and HPV-associated squamous cell carcinoma (SCC) of the uterine cervix is multifactorial. Factors such as a patient’s lifestyle, genetic disposition, and the interplay between HPV genotypes and individual immune responses play important roles [26,27,28,29]. Data from large cohorts have demonstrated that different HPV genotypes vary in their carcinogenic potential [20,30]. HPV16 is associated with 55–60% of all cervical cancers [31,32], and has the highest carcinogenic potential [19,33]. HPV16 is less prevalent in adenocarcinoma (ADC) compared to SCC, with similar trends observed for HPV31, 33, 52, and 58. In contrast, HPV18 and HPV45 are significantly more prevalent in ADC than SCC [31,32].

A study by Kitchener et al. found that women positive for HPV16 had a cumulative incidence of 43.6% for developing CIN2+ over 6 years, compared to 20.1% for any other hr-HPV genotypes [34]. Similarly, research by Sand et al. reported the 8-year absolute risks for CIN3+ with persistent HPV infections as follows: women with HPV16 (55%), HPV33 (33%), HPV18 (32%) HPV31 (31%), and HPV45 (21%) [30,35]. The carcinogenic potential of persistent HPV18 and HPV45 may be underestimated in the studies by Kitchener et al. and Sand et al., as well as in our own research. These genotypes are more likely to affect cylindrical epithelium rather than squamous epithelium, and neither study included these lesions.

Given that each HPV genotype has different oncogenic potential, it is of interest to study the regression rates associated with different genotypes. To date, however, studies have only compared regression rates in women with HPV16-induced premalignant lesions to those with other high-risk genotypes [9,25,36].

In the present prospective study, 34 in 161 women aged 25–40 years with biopsy diagnoses of CIN2-3 showed regression in the excised cone tissue, which was obtained at a mean of 113 days after CIN2-3 diagnosis. HPV35 as the cause of lesions was the only isolated genotype showing a significantly higher rate of regression compared to the other types (Table 1). This is in line with a larger cohort study of 33,288 women screened in Copenhagen, Denmark, where none (0/46) of the women with a single HPV35 infection were diagnosed with CIN3+ over a follow-up period of 11.5 years, and correspondingly, only 8.6% (19/220) of the women with HPV35 infection in combination with other genotypes were diagnosed with CIN3+ [20]. This indicates a less aggressive course of CIN lesions caused by HPV35 in Northern European women. However, this does not apply to women of African ancestry [37,38], emphasizing the need for caution when drawing conclusions from studies based on a single population, as each HPV genotype has varying distribution, oncogenic potential, and prevalence across different regions worldwide [39].

In the present study, when grouping the genotypes with higher and lower regression rates together, the high-regression group (HPV18, 33, 35, 39, and 45) showed a significantly higher regression rate of 35% (21/60) compared to 13.4% (13/97) in the low-regression group (HPV16, 31, 52, and 58) (HR 2.85 versus 0.54). Notably, both HPV18 and 45 clustered in the high-regression group for squamous lesions. According to data from de Sanjosé et al. [32], HPV45 is the third most prevalent genotype in cervical cancer globally, accounting for 6% of cases, following HPV18 (10%) and HPV16 (61%). For squamous cell carcinoma specifically, the genotype distribution is approximately HPV16 (62%), HPV18 (8%), and HPV45 (5%). Data from four Northern countries (Denmark, Norway, Sweden, and Iceland) show similar prevalence patterns for SCC (HPV45 = 7.2%, HPV18 = 10.9%, HPV16 = 59.4%) among women included in the study. They also found that HPV16, 31, and 33 were the most common types in CIN2 and CIN3, while HPV16, 18, and 45 were the most common types in cervical cancer, with HPV18 and HPV45 being far more common in AC than in SCC [40]. While interpreting our findings, it is important to note that HPV18 and HPV45 are more commonly found in ADC than in SCC [36,41,42], and our study did not include lesions of columnar epithelium. This highlights the need for further research into the roles of HPV18 and HPV45 in CIN versus AIS, and their impact on progression and regression.

There are few studies on the regression rates of high-grade CIN lesions in relation to factors such as biomarker expression (pRb, p53), local immune response, and lesion size [8,9,25,36,42,43].

Retinoblastoma protein (pRb) is one of the biomarkers shown to predict CIN lesion behavior. It is a protein encoded by the RB1 tumor suppressor gene. This protein is degraded by the HPV-produced oncoprotein E7, allowing for the replication of cells with damaged DNA [44]. In this study, upon analyzing the whole study population without dividing it into HPV genotype groups, ≤30% of pRb-positive cells in the lower half of the epithelium (pRbLH) indicated a significantly lower probability of regression (HR 0.21, 0.07–0.61). This is consistent with the findings of Baak et al. [45] and is further supported by the results of Kruse et al. who described that a decrease in Rb-positive nuclei in the deeper half of the epithelium was a strong independent predictor of CIN1-2 progression [46]. After genotype stratification, pRbLH was still a predictor for regression in the low-regression group (HR 0.09, 0.01–0.72), but not in the high-regression group. Further investigation into how specific HPV genotypes influence Rb expression is needed for a better understanding of its role.

Product of the tumor suppressor gene TP53, p53 protein, has a pro-apoptotic function, which is often altered in cancers. Mutations and binding to other proteins can affect the potency and stability of p53. The HPV-coded protein E6 binds to and promotes the degradation of p53, potentially reducing p53-induced apoptosis. [47]. Interestingly our data show a higher expression of p53 in the lower half of the epithelium (p53LG > 10%) for lesions with a low regression rate than lesions with a high regression rate. This indicates that the deregulation of p53 subsequently reduces the tumor suppressor effect. This is in line with several studies showing high expression of p53 in cervical cancer [34,48,49]

In clinical practice, the size of the CIN lesion is often evaluated by colposcopy and considered to affect the prognostic outcome. Despite this observation, only a few concrete studies have evaluated the connections among CIN prognosis, HPV genotype, and actual lesion size by diagnosis. Our data show that a CIN2-3 lesion length greater than 2.5 mm in biopsies is associated with a lower regression rate (HR 0.39, 0.19–0.80) (Table 3). Furber et al. found similar correlation for CIN1 lesions size measured by colposcopy. Lesions covering more than 50% of the ectocervix were twice as likely to progress compared to smaller lesions covering less than 25% [50]. One could argue that the increased regression rates among smaller lesions may be due to the fact that punch biopsies remove most of or the whole lesion. However, the difference in regression rates between the high- and low-regression groups (8/34 versus 12/23, Table 3) suggests the involvement of other factors, with studies confirming that immune response, HPV genotype, and the presence of multiple HPV genotypes also play important roles in the regression process [12,48,51,52,53,54].

We have previously shown that CD4-, CD8-, and CD25-positive cells, in addition to tumor suppressors, play roles in the regression of established CIN2-3 [12,13,55]. The activation of CD4+ (T-helper) cells represents the early onset of the adapted immune system [56]. CD4 is a marker for two distinctive groups of T-helper cells: T-helper cell 1 (Th1) and T-helper cell 2 (Th2). Th1 promotes T-cell proliferation and the activation of cytotoxic T cells (CD8+), which are crucial for the clearance of HPV-infected epithelial cells. Th2 cells promote the local humoral response [57], but during a persistent hr-HPV infection, chronic Th2-type inflammation leads to immune suppression [58]. This could explain our findings that a higher stromal ratio of CD8+/CD4+ T cells is favorable for regression, and vice versa. A recent review and meta-analysis of 73 studies by Litwin et al. also found a reduced CD4+/CD8+ ratio in CIN lesions undergoing regression [59].

CD25, also known as interleukin-2 receptor alpha chain, is located on activated B-lymphocytes, but also on regulatory T-cells (T-reg), suppressing the immune response [60,61,62]. A recent review outlining the microenvironment in CIN, as well as regression outcomes, persistence, and response to (immune) therapy, suggested that a low number of T-regs and a higher proportion of CD8+ T cells are favorable for regression. The review also indicated that spontaneous regression is characterized by a higher CD4+/CD25+ ratio, which is in line with our results [22]. The association between a high CD4+/CD25+ ratio and a higher regression rate suggests that CD4+ Th1 activation, along with minimal stimulation of immunosuppressive T-regs, is favorable for lesion disappearance.

The strengths of this study are its prospective design and strict patient inclusion and exclusion criteria, which allowed for a homogenous study population with decreased disturbance of external biases, such as age, health condition, interval between biopsy and cone excision, and previous CIN history. Therefore, there was no significant difference between the regression and non-regression groups regarding mean age. Stavanger University Hospital is the only hospital in the region, ensuring a uniform inclusion strategy, consistent laboratory diagnostics, and minimizing selection and procedure biases. HPV analyses were successfully performed by isolating DNA directly from the actual histology section in 96% (154/161) of the biopsies; this secured the actual HPV infection(s) present in the high-grade CIN lesion, but introduced a potential error, as DNA in paraffin-embedded tissue is at risk of denaturization [63,64]. The limitations of the study include a low number of patients and a relatively short interval between biopsy and cone excision, as regression can take years [65,66]. However, a longer observational period in these patients would have been unethical, as a fraction of patients with high-grade CIN in biopsies already have or will develop cervical cancer. With the introduction of new prognostic biomarkers and strict inclusion criteria, the observational interval may be safely extended. It should be noted that our study focused solely on lesions affecting the squamous epithelium, excluding any dysplasia in the columnar epithelium.

Thanks to new methods, current knowledge about the underlying mechanisms of CIN lesions development, the role of immune system, and the prognostic drivers is increasing. A recent study by our group identified a six-gene signature that can predict CIN3 regression with high accuracy (AUC = 0.85). In a cervical cancer cohort, a high signature score was linked to better survival (*p* = 0.007) and increased immune cell infiltration and activation (*p* < 0.001) [67]. Another study by our group found that CIN3 lesions, compared to normal cervical tissue, had higher expression of oncogenic genes and genes linked to proliferation, T-cell activation, regulation, and differentiation [67]. Other studies have identified DNA targets, like FAM19A4 and miR124-2, with up to 98.3% methylation in cervical cancer cohorts [68,69]. In line with this, a recent study following 114 women with untreated CIN2 or CIN3 for 24 months found a 65.8% regression rate, which was associated with negative FAM19A4/miR124-2 methylation [70]. Emerging technologies for the spatial mapping of mRNA and protein expression will help to further refine our understanding of premalignant cells and their microenvironment. This can optimize the risk-stratification of CIN lesions and improve women’s health by reducing unnecessary follow-up for low-grade lesions, preventing the overtreatment of high-grade lesions that are likely to regress, and ensuring efficient care for those at high risk of cancer progression. New prognostic biomarkers will be of great benefit in improving the management of the vaccinated population with less aggressive HPV genotypes.

## 4. Materials and Methods

### 4.1. Study Population

This study was approved by the Norwegian Regional Ethical committee (#NR303.06, 2012/1292), the Social and Health Department (#07/3300) and the Norwegian Social Science Data Service (#17185). Written informed consent was obtained from all patients for inclusion in the study.

Two-hundred and fifty-four women aged 24–40 years who had been referred to the gynecology outpatient clinic for histological examination at Stavanger University Hospital between January 2007 and December 2008 were prospectively included. According to previous Norwegian Guidelines (2005) [71], the cytology diagnoses prior to inclusion were as follows: (1) low-grade squamous intraepithelial lesion (LSIL) and high-risk HPV (hrHPV)-positive, (2) atypical squamous cells of undetermined significance (ASCUS) and hrHPV-positive, (3) at least one year of persistent hrHPV infection, (4) atypical squamous cells cannot rule out a high-grade lesion (ASC-H), or (5) high-grade squamous intraepithelial lesion (HSIL). Women with cytology diagnoses of atypical glandular cells (AGUS) and adenocarcinoma in situ (AIS) were not included. None of the included women were vaccinated, as organized HPV vaccinations for girls in Norway started in 2009 and the catch-up vaccination for women aged 18–26 years old took place in 2016–2018. This study‘s exclusion criteria were histological biopsy results other than CIN2-3 (normal *n* = 35; CIN1 *n* = 14; adenocarcinoma in situ *n* = 1), pregnancy during the inclusion period (*n* = 11), diseases affecting the immune system or the need for immunosuppressive treatment (*n* = 9), previous treatment for CIN (*n* = 7), a biopsy–cone interval shorter than 80 days (*n* = 7), and laboratory technical issues (insufficient lesions for immunohistochemistry or low DNA quality) (*n* = 9) (Figure 1). The final study population consisted of one hundred and sixty-one healthy women with first-time onset of CIN2-3 (*n* = 161) in colposcopy-directed biopsies. Regression was defined as CIN1 or less in the subsequent cone excision (*n* = 34). Persistence was defined as having CIN2-3 in both biopsy and cone tissue. The mean age was calculated for the study population as a whole and separately for the regression and persistent groups. Except for one patient, this study population was identical to that of a former publication also presenting the results of CD4+ cells in stroma, pRbLH, and CIN lesion length [12].

### 4.2. Gynecologic Methods

A general gynecologic examination including colposcopy-directed biopsies and endocervical curettage was performed. If the biopsies diagnosed CIN2 or CIN3, cone excision with a LEEP was performed after a mean time of 113 days (range: 84–171), reflecting the standard waiting time between biopsy and cone excision in many Norwegian hospitals and comparable with intervals used by others [14]. A LEEP performed more than 16 weeks after punch biopsy was related to a delay on the patient ‘s part, not the study procedure.

### 4.3. Pathology

According to the standard operating procedures, the punch biopsies and LEEP material were fixed in 4% buffered formaldehyde (24–48 h) at 20 °C and embedded in paraffin at 56 °C. Standard hematoxylin–eosin–saffron (HES) sections were used for histological evaluation and independently reviewed by two experienced gynecological pathologists, blinded against each other’s diagnoses. The proliferation marker Ki67 (nuclei staining) and the tumor suppressor protein p16 (cytoplasmatic staining) were used to optimize the diagnoses [55,72].

### 4.4. Semi-Quantitative Scoring of Immunohistochemical Staining and CIN Length Measurements

Immunohistochemistry was performed according to the protocol, as previously described in [55], with staining antibodies targeting p16 (clone E6H4, CINtec Histology kit, ready to use; MTM laboratories, Heidelberg, Germany); Ki67 (clone MIB-1, 1:100; DAKO, Glostrup, Denmark); retinoblastoma protein (pRb) (clone 13A10, 1:25; Novocastra, Newcastle upon Tyne, UK); p53 (clone DO-7, 1:200; DAKO, Glostrup, Denmark); and the immune markers for CD4 (clone 1F6, 1:20; Novocastra, Newcastle upon Tyne, UK), CD25 (clone 4C9, 1:150; Novocastra, Newcastle upon Tyne, UK), CD8 (clone C8/144B, 1:50; DAKO, Glostrup, Denmark), and CD138 (clone B-B4, 1:200; Serotec, Kidlington, UK) [55]. The consensus scoring of staining in the most severely dysplastic areas with the most intensive Ki67 and p16 staining was conducted by three observers (J.P.A.B., E.G.G., and A.C.M.), and independently quality-controlled by a fourth observer (I.T.Ø). All observers used the same microscope and a 40× objective field of vision (0.52 mm, numerical aperture of 0.65). To ensure the presence of the same CIN lesion in all sections, a section adjacent to the sections used for immunohistochemical staining was cut and stained with HES, p16, and Ki67 (“sandwich technique”).

For pRb and p53, the epithelium in one field of vision was divided into an upper (pRbUH or p53UH) and a lower half (pRbLH or p53LH), and the percentage of positive and negative cells in each layer was calculated. For CD8 and CD25, all positive cells in the epithelium were counted in two fields of vision; for CD4, only the nucleated positive cells with a lymphocytic morphology were counted. In the stroma, adjacent to the already scored epithelium, all cells positive for CD138, CD8, CD4, and CD25 were counted in two fields of vision.

The CIN lesion length was measured in the representative HES section of all punch biopsies. If a punch biopsy contained CIN, the length was assessed using the diameter of the field of vision (i.e., 0.52 mm). If more than one biopsy contained CIN, the length of each lesion in the actual biopsies was summarized. In randomly selected cases comprising 10% of the total cases, quality control of the length measurements was performed using the QPRODIT (version 6.1) interactive image analysis system (Leica, Cambridge, UK), as previously described by Kruse et al. [72]. The correlation between the microscopic and QPRODIT measurements was high (R = 0.99, *p* < 0.0001).

### 4.5. HPV Analyses

DNA material from all formalin-fixed, paraffin-embedded (FFPE) biopsies at inclusion was isolated (E.Z.N.A.^TM^ Tissue DNA Kit, Omega Bio-Tek, Inc. Norcross, GA, USA). HPV analyses were performed using the Roche Linear Array (LA) HPV Genotyping test (Roche Molecular Systems, Roche Diagnostics, Mannheim, Germany). The LA primers amplified HPV-DNA from 37 HPV genotypes (6, 11, 16, 18, 26, 31, 33, 35, 39, 40, 42, 45, 51, 52, 53, 54, 55, 56, 58, 59, 61, 62, 64, 66, 67, 68, 69, 70, 71, 72, 73, 81, 82, 83, 84, 89, and IS39 (subtype of 82)), in addition to β-globin DNA as a cellular control [55].

The INNO-LiPA *Extra* II kit (IL) and the TENDIGO machine (Fujirebio Europe N.V., Gent, Belgium) were used to genotype 11 cases with invalid results in the LA assay. The IL detected 32 different genotypes (6, 11, 16, 18, 26, 31, 33, 35, 39, 40, 42, 43, 44, 45, 51, 52, 53, 54, 56, 58, 59, 61, 62, 66, 67, 68, 70, 73, 81, 82, 83, and 89), in addition to the HLA-DPB1 gene as a DNA control. These genotypes were also detected via LA. For both genotyping tests, DNA was amplified using multiplex PCR with pooled biotinylated HPV and human DNA control primers. The amplicons were chemically denatured, and the samples were hybridized to strips coated with HPV probes representative of the genotypes detected by each method, in addition to DNA control probe lines. Following the washing steps, the biotin-labeled amplicons that had hybridized to the complementary oligonucleotide probes on the strip were visualized in the presence of conjugate and substrate solutions. For the LA assay, reagents were added and removed manually using an automated pipette and vacuum aspirator, while the hybridization and subsequent washing and color development steps were fully automated in the IL protocol. Two independent readers interpreted the results; any discrepancies were referred to a third reader [30].

For six of the cases, neither LA nor INNO-LiPA provided successful DNA genotyping from the biopsies. Genotyping was, therefore, performed via LA using DNA from the ThinPrep liquid-based cytology (LBC), sampled at a mean of 48 days (range 17–68) ahead of the biopsy, and isolated on the automated MPLC (Roche Diagnostics GmbH, Mannheim Germany), as described by Ovestad et al. [55]. All HPV genotyping was performed after the diagnostic part of the study and had no influence on the follow-up of the patients. Because of the small number of patients with ≤3 regression cases each, HPV6, 42, 53, 54, 56, 59, 61, 66, 68, 73, and 81 were grouped together as “other HPVs”. Patients with more than one HPV genotype were grouped into each of the actual respective groups, i.e., more than one group.

### 4.6. Statistical Analyses

Receiver operating characteristic curve (ROC) analyses were performed to calculate the optimal cut-off values for the highest true-positive and the lowest false-positive rates of the different biomarkers [73]. A time-to-event approach was employed using the time from biopsy to LEEP as the time scale. Parametric accelerated failure time models with exponential distribution were used to estimate the regression rate per 1000 persons by HPV genotype. Cox proportional hazard regression was used to estimate the hazard ratios (HRs) and 95% confidence intervals (CIs) for common HPV genotypes (*n* ≥ 3). Based on the HRs, the HPV genotypes were categorized into two groups: one associated with a higher rate of regression (high-regression group: HPVs 18, 33, 35, 39, and 45) and the other associated with a lower regression rate (low-regression group: HPVs 16, 31, 52, and 58). To examine the possible effects of having multiple HPV infections, we also conducted a sensitivity analysis by excluding those who had more than one infection. The associations among CIN lesion length, biomarkers, and regression overall and for each HPV group were estimated. The discriminative performance of the model was assessed using Harrell’s C, a concordance index measuring the goodness of fit for models producing risk scores. In this study, Harrell’s C describes, as a percentage, how well the actual panel of biomarkers predicts regression. All the tests were two-sided, and the results were considered significant at *p* < 0.05. Stata Version 16 was used for all the analyses.

## 5. Conclusions

In conclusion, we found that the size, molecular pattern of cell cycle regulators, and immune cell response differed in regression compared to persistent high-grade CIN lesions. Only HPV35 was associated with a significantly higher regression rate, while other genotypes showed non-significant tendencies. Comparing the group with a higher tendency for regression to the one with a lower tendency revealed significant differences in the ratios of immune cells. This study has enhanced our understanding of how different HPV genotypes and genotype groups, in relation to a panel consisting of different tumor suppressors, immune cell constitutions, and CIN lesion sizes, can predict CIN2-3 outcomes. This provides valuable insights into virus–host interactions. However, a deeper understanding of the molecular drivers in CIN and cervical cancer is essential for improving risk stratification and developing new therapeutic strategies that could be utilized in future conservative treatments for some CIN2-3 patients. Identifying biomarkers in the development of high-grade CIN lesions will enable optimized recommendations, ensuring adequate treatment and follow-up for women at high risk of cancer. Additionally, it could prevent unnecessary treatment for women at low risk of cancer and with a high probability of regression. This is particularly important as the proportion of vaccinated women with less dangerous lesions or HPV genotypes increases. Therefore, further research into robust prognostic markers or panels should be conducted in larger, independent cohorts of high-grade CIN and cervical cancer patients.

## Figures and Tables

**Figure 1 ijms-26-05205-f001:**
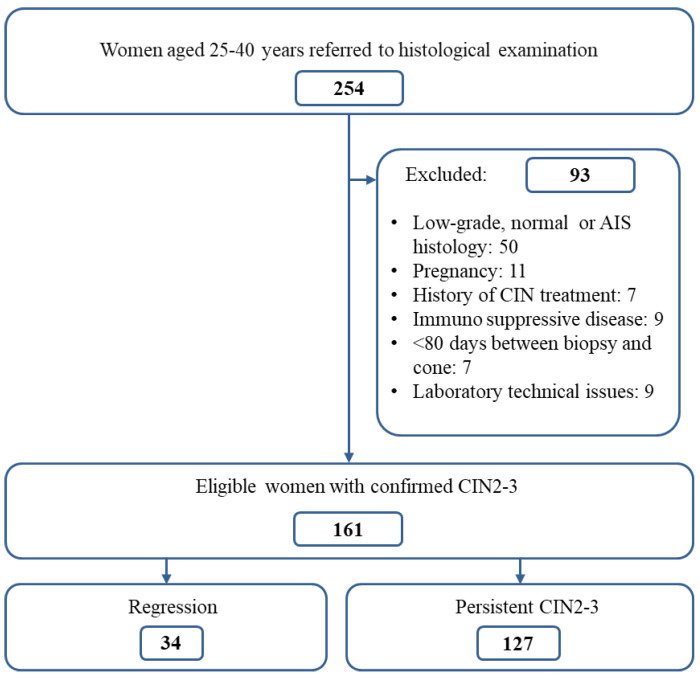
Overview of inclusion and exclusion criteria. Flowchart showing actual number of referred patients at study start, number of patients excluded by defined criteria, remaining study population with histologically proven CIN2-3, and distribution of women with regression and persistent CIN2-3.

**Table 1 ijms-26-05205-t001:** Regression status (yes or no), regression rate per 1000 persons, and hazard ratio (HR) according to HPV genotype.

HPV Genotypes (Yes/No)	Regression	Regression Rate per 1000	HR (95% CI)
**HPV16**			
No	26/76	2.26 (1.50–3.02)	1.00
Yes	8/51	1.20 (0.42–1.97)	0.53 (0.26–1.10)
**HPV31**			
No	31/102	2.10 (1.42–2.70)	1.00
Yes	3/25	0.96 (0.00–1.99)	0.46 (0.15–1.43)
**HPV52**			
No	32/116	1.92 (1.33–2.51)	1.00
Yes	2/11	1.31 (0.00–3.03)	0.69 (0.18–2.60)
**HPV58**			
No	34/119	-	-
Yes	0/8	-	-
**HPV18**			
No	29/118	1.74 (1.17–2.31)	1.00
Yes	5/9	3.35 (1.05–5.64)	1.93 (0.90–4.13)
**HPV33**			
No	28/114	1.74 (1.16–2.32)	1.00
Yes	6/13	2.89 (0.93–4.85)	1.66 (0.78–3.54)
**HPV35**			
No	28/120	1.67 (1.11–2.24)	1.00
Yes	6/7	4.15 (1.74–6.57)	2.48 (1.27–4.86)
**HPV39**			
No	31/120	1.81 (1.24–2.39)	1.00
Yes	3/7	2.72 (1.91–5.25)	1.50 (0.56–4.00)
**HPV45**			
No	31/119	1.83 (1.25–2.41)	1.00
Yes	3/8	2.45 (0.06–4.85)	1.34 (0.48–3.74)

**Table 2 ijms-26-05205-t002:** Regression status (yes or no), regression rate per 1000 persons, and hazard ratio (HR) according to low-regression group and high-regression group.

HPV Genotypes (Yes/No)	Regression	Regression Rate per 1000	HR (95% CI)
**Low-regression group ^1^**			
No	21/43	2.92 (1.89–3.95)	1.00
Yes	13/84	1.18 (0.58–1.78)	0.54 (0.22–0.75)
**High-regression group ^2^**			
No	13/88	1.11 (0.55–1.68)	1.00
Yes	21/39	3.20 (2.09–4.30)	2.85 (1.54–5.28)

^1^ HPVs 16, 31, 52, and 58. ^2^ HPVs 18, 33, 35, 39, and 45.

**Table 3 ijms-26-05205-t003:** Regression (yes/no) for the total study population and for the high-regression and low-regression groups.

	Total		High-Regression Group ^1^		Low-Regression Group ^2^	
Variables	Regression (yes/no)	HR ^3^ (95% CI)	Regression (yes/no)	HR ^3^ (95% CI)	Regression (yes/no)	HR ^3^ (95% CI)
pRb+ in lower half of epithelium						
≤30%	4/58	0.21 (0.07–0.61)	2/16	0.21 (0.04–1.14)	1/45	0.09 (0.01–0.72)
>30%	30/68	1.00	19/23	1.00	12/38	1.00
Length of CIN lesion						
≤2.5 mm	22/40	1.00	13/10	1.00	8/26	1.00
>2.5 mm	12/87	0.39 (0.19–0.80)	8/29	0.48 (0.18–1.26)	5/58	0.31 (0.91–1.04)
p53 LH+ in lower half of epithelium						
≤10	23/58	1.00	13/18	1.00	9/41	1.00
>10	11/68	0.32 (0.15–0.71)	8/21	0.84 (0.35–1.99)	4/42	0.20 (0.04–0.88)
CD4/CD25 in stroma						
≤9.75	26/115	1.00	17/35	1.00	10/72	1.00
>9.75	7/11	4.96 (2.46–9.99)	3/4	4.27 (1.22–14.87)	3/11	6.17 (1.29–29.57)
CD4/CD8 in stroma						
≤0.67	27/81	1.00	17/21	1.00	10/50	1.00
>0.67	6/46	0.43 (0.21–0.91)	3/14	0.34 (0.11–1.05)	3/34	0.36 (0.11–1.17)
Harrell’s C ^4^		78%		71%		82%

Regression is defined as normal histological scoring or CIN1 in the cone biopsy. ^1^ HPVs 18, 45, 33, 35, and 39. ^2^ HPVs 16, 31, 52, and 58. ^3^ Hazard ratios (HRs) and 95% confidence intervals (CIs) were estimated using a Cox proportional hazards ratio model. ^4^ The percentage that the model correctly predicted.

## Data Availability

The datasets presented in this article are not readily available because the study is based on participant consent, and any access to the data must comply with the conditions outlined in the informed consent documentation. Requests to access the datasets should be directed to ane.cecilie.munk@sshf.no.

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
