# Peer review of "The Predictive Impact of HPV Genotypes, Tumor Suppressors and Local Immune Response in the Regression of Cervical Intraepithelial Neoplasia 2-3: A Prospective Population-Based Cohort Study"

_ijms, 2025, doi:10.3390/ijms26115205_

Round 1

Reviewer 1 Report

Comments and Suggestions for Authors

Dear Authors,

I would like to commend you on your hard work and thorough research, which makes a valuable contribution to the field of HPV-related studies. The manuscript is well-structured, clearly outlining the study population, study design, treatments, and analytical methods. The results are presented with precision, and the discussion offers clear interpretations of the data. The conclusion is well-supported by the findings and demonstrates a solid understanding of the subject matter.

However, there are a few details that need to be addressed before publication.

Abstract, line 22: The sentence "Cervical intraepithelial neoplasia (CIN) is caused by human papillomavirus (HPV)" is mostly accurate but overly simplistic. Please consider modifying it slightly to reflect that HPV is a necessary but not sufficient cause of CIN—that is, while HPV infection is required for CIN to develop, it does not always lead to neoplasia on its own. A small addition of a few words would be enough to clarify this important detail.

Line 218: The phrase "It is a tumor suppressor gene which is degraded by the HPV-produced..." could benefit from a brief clarification. Consider specifying that the protein is encoded by the RB1 tumor suppressor gene.

Author Response

1. Summary

Dear Reviewer,

Thank you for your kind words and positive feedback on our manuscript. We are pleased to hear that you found our work well-structured and valuable to the field of HPV-related studies. We appreciate your thorough review and will address the details you mentioned before the publication.

3. Point-by-point response to Comments and Suggestions for Authors

Comments 1: Abstract, line 22: The sentence "Cervical intraepithelial neoplasia (CIN) is caused by human papillomavirus (HPV)" is mostly accurate but overly simplistic. Please consider modifying it slightly to reflect that HPV is a necessary but not sufficient cause of CIN—that is, while HPV infection is required for CIN to develop, it does not always lead to neoplasia on its own. A small addition of a few words would be enough to clarify this important detail.

Response 1: Thank you for pointing this out. We totally agree with this comment. We will change the sentence to: "Cervical intraepithelial neoplasia (CIN) is caused by human papillomavirus (HPV); however, factors such as HPV genotype and individual immune response may also contribute to its development."

Comments 2: Line 218: The phrase "It is a tumor suppressor gene which is degraded by the HPV-produced..." could benefit from a brief clarification. Consider specifying that the protein is encoded by the RB1 tumor suppressor gene.

Response 2: Thank you for noticing this. We agree with this comment. We will change it to: “Retinoblastoma protein (pRb) is one of the biomarkers shown to predict CIN lesion behavior. It is a protein encoded by the RB1 tumor suppressor gene. This protein is degraded by the HPV-produced oncoprotein E7, allowing replication of cells with damaged DNA”

Thank you for your feedback. Please let us know if you have any further comments or suggestions.

Best regards

Dr. Pavla Sustova

Reviewer 2 Report

Comments and Suggestions for Authors
  1. Congrats to the authors for this very interesting and important piece of work. The work has been well presented.
  2. I am not sure why the authors consider this study a population-based cohort study. The term population-based gives the impression that the participants were randomly selected from the general population but this was not the case.
  3. The word "grade" can be removed from the title
  4. There is a mixed up with the arrangement; material and methods should come before results and discussion
  5. The authors have used the terms conization and LEEP interchangeably. These 2 are not the same. It appears the authors did LEEPs, not cone. This has to be rectified

Author Response

1. Summary

Thank you very much for taking the time to review this manuscript. Please find detailed responses below and the corresponding revisions/corrections highlighted/in track changes in the re-submitted files.

3. Point-by-point response to Comments and Suggestions for Authors

Comments 1: Congrats to the authors for this very interesting and important piece of work. The work has been well presented.

Response 1: Thank you for your positive feedback and kind words. We are pleased to hear that you found our work interesting and well presented. We appreciate your support and look forward to any further comments or suggestions you may have.

Comments 2: I am not sure why the authors consider this study a population-based cohort study. The term population-based gives the impression that the participants were randomly selected from the general population, but this was not the case.

Response 2: Thank you for pointing this out. We consider this study to be a population-based cohort study, because it involves individuals from a defined general population over time assessing associations between HPV exposure, CIN development and health outcomes (regression/persistence). While the term "population-based" might suggest random selection from the general population, it is important to note that "population-based" can also refer to studies that include all individuals within a defined population, regardless of the selection method. In our case, the study design aimed to capture a representative sample of the population, even if the selection was not random (all women between 25-40 years of age referred to biopsy according to the guidelines were invited to participate in the study). The key aspect is that the cohort is intended to reflect the broader population from which it was drawn, allowing for generalizable findings. Therefore, we believe it is appropriate to refer to this as a population-based cohort study.

Comments 3: The word "grade" can be removed from the title

Response 3: We agree with this comment. Therefore, we have removed the term "grade" from the title.

Comments 4: There is a mixed up with the arrangement; material and methods should come before results and discussion

Response 4: We understand that the order is unusual, but we have followed the manuscript template: ( https://urldefense.proofpoint.com/v2/url?u=http-3A__www.mdpi.com_files_word-2Dtemplates_ijms-2Dtemplate.dot&d=DwIF-g&c=euGZstcaTDllvimEN8b7jXrwqOf-v5A_CdpgnVfiiMM&r=bhgXBxYpHDjsn6zH9ADc1RouTxQUDnzmT0zBS-l8K1s&m=WeE63-zaow1dNU3zPIB5yzL8vFYD1pbLS1N8Owa1rJJ7mKXTeota-fED9e5_oc2L&s=lPfDhkOKlfVJcppUBctWKveeH5pIRsxzwKeftYBPQ4g&e=), which specifies that the results and discussion should precede the material and methods section. We have confirmed that this is the correct order with Ms. Demi Dong, our Section Managing Editor.

Comments 5: The authors have used the terms conization and LEEP interchangeably. These 2 are not the same. It appears the authors did LEEPs, not cone. This has to be rectified

Response 5: We agree with this comment and will ensure that LEEP is used consistently throughout the article.

Thank you for your feedback. Please let us know if you have any further comments or suggestions.

Best regards

Dr. Pavla Sustova